# Botulism in Spain: Epidemiology and Outcomes of Antitoxin Treatment, 1997–2019

**DOI:** 10.3390/toxins15010002

**Published:** 2022-12-20

**Authors:** Marina Peñuelas, María Guerrero-Vadillo, Sylvia Valdezate, María Jesús Zamora, Inmaculada Leon-Gomez, Ángeles Flores-Cuéllar, Gema Carrasco, Oliva Díaz-García, Carmen Varela

**Affiliations:** 1Escuela Internacional de Doctorado, Universidad Nacional de Educación a Distancia (UNED), Calle de Bravo Murillo, 38, 28015 Madrid, Spain; 2Department of Communicable Diseases, National Centre of Epidemiology, Instituto de Salud Carlos III, C/Monforte de Lemos 5, Pabellón 12, 28029 Madrid, Spain; 3Laboratorio de Referencia e Investigación en Taxonomía, Bacteriología, Centro Nacional de Microbiología, Instituto de Salud Carlos III, Carretera Pozuelo-Majadahonda Km 2.2, 28220 Madrid, Spain; 4Servicio de Microbiología Alimentaria, Centro Nacional de Alimentación, Agencia Española de Seguridad Alimentaria y Nutrición, Ctra. Pozuelo a Majadahonda Km 5.1, 28220 Madrid, Spain; 5Consortium for Biomedical Research in Epidemiology and Public Health (CIBERESP), 28029 Madrid, Spain; 6Medicines for Human Use Department, Agencia Española de Medicamentos y Productos Sanitarios (AEMPS), C/Campezo 1, Edificio 8, 28022 Madrid, Spain

**Keywords:** botulism, food-borne botulism, infant botulism, botulinum neurotoxins, botulism antitoxin

## Abstract

Background: Botulism is a low incidence but potentially fatal infectious disease caused by neurotoxins produced mainly by *Clostridium botulinum*. There are different routes of acquisition, food-borne and infant/intestinal being the most frequent presentation, and antitoxin is the treatment of choice in all cases. In Spain, botulism is under surveillance, and case reporting is mandatory. Methods: This retrospective study attempts to provide a more complete picture of the epidemiology of botulism in Spain from 1997 to 2019 and an assessment of the treatment, including the relationship between a delay in antitoxin administration and the length of hospitalization using the Cox proportional hazards test and Kruskal–Wallis test, and an approach to the frequency of adverse events, issues for which no previous national data have been published. Results: Eight of the 44 outbreaks were associated with contaminated commercial foods involving ≤7 cases/outbreak; preserved vegetables were the main source of infection, followed by fish products; early antitoxin administration significantly reduces the hospital stay, and adverse reactions to the antitoxin affect around 3% of treated cases.

## 1. Introduction

Botulism is an infectious disease caused by the activity of the neurotoxin produced mainly by the anaerobic, Gram-positive and spore-forming bacterium *Clostridium botulinum*, which can be found in the environment, such as the soil and dust. These exotoxins consist of two protomers (an active part in the light chain and a host cell binding part in the heavy chain) linked by a disulfide bridge [1,2]. Botulinum toxins bind to the presynaptic membrane by a double receptor mechanism [3], blocking acetylcholine release at the neuromuscular synapse, causing a progressive symmetrical and descending flaccid paralysis that can be fatal by asphyxia due to diaphragmatic arrest [4]. To date, several different isoforms, grouped into seven toxin serotypes, have been described; from A to G, but only A, B, E and, less frequently, F can cause human botulism [5]. In addition, some new chimeric toxins (C/D, D/C and F5A) have been described recently [6,7]. Although *C. botulinum* is the most common species producing botulinum toxins, toxin type E is also released by *C. butyricum* and toxin type F is usually related to *C. baratii* [8,9]. In addition, neurotoxin-like botulinum has been found in *Chryseobacterium piperi*, *Weisella oryzae* and *Enterococcus* spp. strains [10,11,12,13,14]. Botulism can be acquired by ingestion of improperly processed food containing pre-formed toxin (food-borne botulism); by toxin release at the intestinal level when microbiome and/or intestinal peristalsis cannot prevent spore germination and gut colonization, as happens mainly in children below 1 year of age (infant or intestinal botulism); by contamination of wounds by spores which develop to vegetative forms and then produce the toxin (wound botulism); or, less frequently, due to iatrogenic injection [15] or bioterrorism. Severe cases of all ages usually require ventilation for respiratory failure and admission to the Intensive Care Unit (ICU). Food-borne botulism is found worldwide, related to canned/preserved food, especially home-canned food [16,17], in which the anaerobic environment allows *C. botulinum* to develop. Since *C. botulinum* does not grow in food products with high acidity (pH < 4.6) and/or osmolarity, and the toxin is susceptible to inactivation by heat, foodstuffs that are preserved with low acidity or low osmolarity and consumed without heat treatment are more likely to be associated with botulism [16]. Regarding infant botulism, its relation to honey, where spores can be preserved, is well documented, but there are several possible sources, such as dust [17]. Consequently, national and regional public health primary prevention actions aim to reduce these known risk factors by promoting good practices during home-canning, reducing the production of home-canned food that is not reheated before consumption, and avoiding the consumption or use of honey by children under 1 year of age [18,19,20]. Feces, serum and/or gastric juice are required for laboratory diagnosis; taking into account that toxin detection in serum is less probable than in feces, and that the serum sample must be collected before antitoxin administration [21,22].

The treatment of choice consists of the administration of antitoxin as soon as possible. Different equine antitoxin formulations including two (AB) or three (ABE, available from 2008 to 2018) serotypes have been used, and, since 2018, despeciated heptavalent Botulism Antitoxin (BAT) [23,24], which includes all serotypes, is used in Spanish hospitals for patients of all ages. From 2007, human antitoxin including serotypes A and B (BabyBIG), which has a longer half-life and requires a lower dose than BAT, has been used as the treatment of choice for infant botulism [25,26].

According to the latest ECDC annual report [27], with data from 2015, and the ECDC Atlas with 2021 data [28], botulism in Europe has remained stable over the last decade. The reporting rate was <0.1 cases per 100,000 population, with children under 1 year of age being the group with the highest rate. Time series analysis with data between 2011 and 2015 did not show a clear pattern. In Spain, national notification of all possible botulism cases is mandatory within the National Epidemiological Surveillance Network (RENAVE, by its acronym in Spanish), and, if an outbreak is suspected, or the suspected product is commercialized, or intentional contamination is suspected, the notification at national level must be done urgently [29]. Case information is stored on an electronic platform managed by the National Centre for Epidemiology.

This study aims to answer questions related to the epidemiological characteristics of botulism in Spain, including those associated with the outcomes of antitoxin administration, in order to increase early medical investigation, improve patient management and review the main risk factors identified and its trend to reduce or avoid them.

## 2. Results

### 2.1. Epidemiological Description of Cases

The analysis was carried out on 232 cases that met the case definition (Table 1) and found 92 (39.66%) possible, 63 (27.16%) probable and 77 (33.19%) confirmed cases (Table 2). Within the 63 probable cases, laboratory evidence of botulinum toxin in food was found in 68.25% of the cases, while the remaining 31.75% were associated with eating the same food of a confirmed case. There were 204 (87.93%) food-borne cases and 28 (12.07%) intestinal botulisms, with no wound-related infections. The overall notification rate was 0.02 cases per 100,000 inhabitants/year. The median age in the food-borne cases was 51 years (IQR = 31, range 0–85 years), of whom 55.88% were male, without differences based on sex (*p*-value = 0.144). No woman was reported to be pregnant at the time of the event. The median age of the infants with botulism was 3 months (IQR = 5, range 20 days–10 months), with 57.14% male, and without differences based on sex (*p*-value = 0.214). According to the hospital registers, there were nine cases with ICD codes for alternative diagnostics: four cases of Guillain–Barré, two myasthenia gravis and three related to stroke. Seven of these nine cases were classified as possible, with botulism being the main diagnosis in six of them (for the other one it was Guillain–Barré), and two were confirmed as botulism after hospital discharge.

Among the 204 cases of food-borne botulism, 57.35% (117 cases) belong to one of the 44 **outbreaks** notified during this period (Appendix A). The annual mean and median number of outbreaks was 1.9 and 2 respectively, with a huge incidence in the summer quarter of the year (51.75% of outbreak-related cases) (Figure 1). As expected, there were no infant botulism outbreaks. Thirty outbreaks (68.18%) affected two people each, and the median number of cases per outbreak was two, with up to seven cases in two outbreaks. There were 37 outbreaks with information about the most likely setting of the exposure, home being the most frequent (33 outbreaks), away from home at a restaurant (2 outbreaks) and collective canteens (2 outbreaks). In 25 outbreaks, it was possible to recover food samples, with 72% positives results (Appendix A). The two toxin type E cases were included in a multi-country outbreak related to salted and dried fish (*Rutilus rutilus*) from Eastern Europe, in which the toxin was detected. The outbreak related to toxin type F, produced by *C. baratii*, caused the ICU admission of an entire family after sharing a home meal, but the source of contamination could not be established [31].

The remaining 115 **sporadic cases** consisted of 87 food-borne botulism (75.65%) and 28 infant botulism (24.35%). Both food-borne and infant botulism cases were more frequent during the summer months (Figure 1). The time until hospital admission and length of hospitalization for both the food-borne and infant botulism cases are shown in Table 3.

Among the **sporadic food-borne cases**, 26 (29.88%) could be confirmed; four (4.60%) were probable cases; and 57 (65.52%) cases were classified as possible. Toxin type B was detected in 46.15% of the confirmed cases; one case was associated with toxin type A; and, for the rest, the toxin type was not available. The source of the toxin was investigated in 41.38% of the sporadic food-borne cases, with 13.88% of the positive results (three toxin type B, one toxin type A and one unspecified), being from homemade canned vegetables. There were no differences in terms of the incubation period among the food-borne cases, related or not to an outbreak, with a mean and median of one day (SD = 1.19; IQR = 2) and five days as the longest incubation period. There were also no differences between the onset of symptoms and hospital admission between the sporadic and outbreak-related cases.

Figure 2 shows the distribution of the toxin frequency by case classification and category (regardless of whether they were part of an outbreak or not), with toxin type B being the most frequent, by far, with respect to all the other types.

To analyze the origin of the most likely food associated with the event, we used all the sporadic food-borne cases and a representative of each outbreak. As a result, 77.78% of these events were linked to homemade preserved food, and the remaining 22.22% were related to trade foodstuffs. Canned vegetables were the most suspected source in 68% of the events, mostly home-preserved (61 out of 68 events), followed by fish products, with two outbreaks related to home-preserved, and six events linked to commercial foods (two outbreaks and four sporadic cases). There were 20 events related to large-scale commercial foods, with fish products (six events) and vegetables (five events) being the most frequently implicated food, but the toxin was detected in only five of those events: two events related to vegetables, two related to fish products and one related to meat.

Figure 3 shows the frequency of food-borne cases by detection of the toxins in food, the food category and trade-type among all the cases (possible, probable and confirmed). It must be noted that, when a commercial food was involved, only one item was found to be contaminated; no other products analyzed from the same batch were affected and no other cases related to the same product were detected.

Regarding **infant botulism**, there were 23 confirmed cases, one probable and four possible cases. Toxin type B was detected in all the cases with information about the serotype (Figure 2). Different samples, mostly food but also an enema containing honey, from 12 cases were analyzed, finding only one positive result for the local commercial honey (toxin type B). Nine of these cases (32%) were exclusively breastfed, and 19 (67.86%) had consumed something else, with herbal tea (seven cases), honey (five cases) and infant formula (five cases) being the most frequent. In addition, in three cases, the families reported living or working next to construction sites, quarries or apiaries. There have been no suspected honey-related cases since 2010.

The severity criteria were detected in 48.53% of the **food-borne cases** and 32.14% of the **infant botulism cases**. No relationship with any analyzed factor was found using logistic regression in the food-borne cases, but, in the infant cases, antitoxin administration was associated with the more severe cases (*p*-value = 0.011). There were eight deaths (3.57% fatality rate, with eight more cases without outcome data) before 2014, with an average age of 67 years (SD = 25.03; median = 68, IQR = 38; range = 35–79 years). Except for one case of a very long hospitalization course (446 days), the mean and median duration of hospitalization in the fatal cases were 11 and 7 days, respectively (SD = 10.25; IQR = 7; range 1–29), being under the overall mean and median of length of hospitalization in the food-borne non-fatal cases (mean 23.61 (SD = 43.95) and median 14 (IQR = 18)).

### 2.2. Antitoxin Treatment

Information related to antitoxin administration was available for 191 (82.33%) of the total cases: 164 food-borne and 27 infant cases (Figure 4). In the food-borne cases, no differences according to the antitoxin treatment group were observed in the severity (*p*-value = 0.817) or in the classification of the case (*p*-value = 0.475).

For the Cox analysis (Table 4), 88 cases out of 103 cases of **food-borne botulism** with information related to the time of antitoxin administration and length of hospitalization were used, while the rest were censored. Adjusting by the antitoxin administration group, sex, age and case classification, the probability of obtaining discharge in a given time is 55.40% lower if the patient received antitoxin >48 h after the onset of symptoms (*p*-value = 0.026), and 73.64% lower if the patient did not receive antitoxin (*p*-value = 0.002), with respect to the early treated. The probability of discharge in a given time decreases 1.18% for each year of life (*p*-value = 0.022). In addition, the probability of discharge in a given time is 50.54% lower for the confirmed cases than the possible cases (*p*-value = 0.014), without a difference between the possible and probable cases. There were no differences based on sex (*p*-value = 0.108). The median length of hospitalization was 10 days (IQR = 10) for those who received antitoxin early, 16 days (IQR = 19) for those who received it later, and 17 days (IQR = 14) for those who did not receive antitoxin treatment (Kruskal–Wallis *p*-value = 0.0513), being statistically significant when comparing between early and late administration (*p*-value = 0.0126). We could not find statistical differences between the late administration of antitoxin and no treatment (Kruskal–Wallis *p*-value = 0.1007).

According to the established criteria, four cases with a possible acute adverse reaction to antitoxin were detected: three males of 58, 67 and 72 years old and a 58-year-old female (Appendix A). All of them took place before BAT commercialization and were likely related to ABE antitoxin.

Seven cases of infant botulism were treated with antitoxin (Appendix A), corresponding to 30.43% of all the infant cases from 2007. All of them were treated >48 h after the onset of symptoms, and five were severe cases. There were no statistical differences between those who received antitoxin and those who did not in terms of length of hospitalization (Kruskal–Wallis *p*-value = 0.1007). Statistical analyses between the two antitoxin types were not performed due to the low number of cases in each group. No adverse events to the antitoxin were found among the infant cases.

## 3. Discussion

The epidemiological data show similarities with those previously published for the European Region [27] in terms of the notification rate, <0.1 cases per 100,000 population. Similarly, the fatality rate (3.57%) was found to be close to the data from countries with similar healthcare systems, such as Italy’s 4% in a series between 1986 and 2015 [21], or around 5% in the US, where type A toxin is more frequent [32,33]. This low fatality rate makes it—fortunately—difficult to assess its relationship with the delay in antitoxin administration. Spanish botulism cases are mostly food-borne, similar to what happens in France and Italy [21,34], but contrary to what happens in the US [35]. In our series, most of the cases occurred in males aged 25 to 80 years and females aged 45 to 80 years, and infants <1 year of both sexes, in line with the data observed in the last European report [27] but with differences in the 15 to 24 year age group, in which the data for both sexes were similar in our series.

Homemade preserved foods are the main source of food-borne disease, similarly to the previously published data [36]. However, it is noteworthy that fish products were implicated in almost one-third of the cases in which the most suspected food was commercial and were the commercial foodstuff with the highest number of positive cases.

Though the toxin type was missing in more than 50% of the food-borne cases (typing can only be performed on feces) in Spain, as in Europe in general, the type B neurotoxin predominates [37], followed, by far, by toxin type A. The only outbreak caused by type E neurotoxin was linked to salted fish products [38,39,40], and all the cases in the type F neurotoxin outbreak required ICU admission due to rapid clinical progression [16,31], as seen in the literature previously.

In our series of **infant botulism** cases, the source was found positive only in one case; it was honey, which is the main food item related to infant botulism described in the literature [41,42,43]. The absence of infant botulism cases related to honey from 2010 may show the success of information campaigns carried out by local public health authorities and pediatricians [44]. However, we should note that honey was among the components of an enema that was the main suspected source of infection in a recent case in Spain (out of the study period), and in a previous case included in this paper, which resembles another possible case reported in Malaysia and published recently, in which the author considers the topical application of medical-grade honey on a wound as the most likely cause of infection [45]. In our series, the suspected food item most mentioned was herbal tea, even if *C. botulinum* was not detected in this food vehicle. Due to the nature of herbal tea, the presence of *C. botulinum* spores is not surprising, although this finding is not very frequent, and cross-contamination once the package is opened cannot be excluded easily. Bianco et al. from Argentina detected spores of *C. botulinum* in 7.5% of chamomile samples, being higher in products sold by weight than in tea bags [46]. More studies are needed to determine the role of herbal tea in infant botulism but, for caution, the Spanish Food Safety and Nutrition Agency recommends avoiding its consumption by children under one year of age [18].

According to our results among **food-borne cases**, early antitoxin administration from the onset of symptoms (≤48 h) is associated with a higher probability of a short stay in hospital, while late administration or no antitoxin administration do not differ substantially in terms of the length of hospitalization. It should be noted that IQR in the late administration group is almost double that in the early administration group, so we can expect some kind of benefit in those patients who received antitoxin closer to the cut-off point. As in our case, recent USA studies have shown a relationship between early antitoxin administration and a shorter hospitalization compared to late administration [23,47]. Even if the general recommendation is to use antitoxin in all cases of botulism, the latest CDC review concludes that patients with mild symptoms and no clinical progression, who have not been treated within the first two days, are unlikely to improve with antitoxin administration, particularly after seven days from the onset of symptoms [36], linking the lack of clinical progression to the absence of the toxin in the bloodstream. From an economic perspective, Anderson et al. quantified the mean extra costs derived from the delay in the administration (>2 days) of antitoxin as 2.5 times higher than early administration [48]. Considering that, in our series, more than half of the patients were admitted around 48 h from the onset of symptoms or later, the window of opportunity is very short, and clinicians from emergency wards must be aware of the need to act fast.

It was not possible to develop a specific analysis comparing the length of ICU stays, due to the frequent lack of data on the dates of admission and discharge in these units. However, it should be mentioned that some clinical guidelines used in Spain recommend admission to the ICU in any case with a high suspicion of botulism [49,50]. This could detract from the association between ICU admission and severity since, in some cases, ICU admission responds to the need for constant monitoring for the early detection of symptoms of worsening respiratory function, explaining the lack of differences in the mean and median length of hospitalization between cases with severity criteria and those without them. Furthermore, it is well known that the severity of symptoms is proportional to the dose of toxin [36], and this information exceeds the data routinely collected for case studies, being a major limitation.

Regarding the classification of cases, the low level of confirmed cases may be associated with the limitations of the bioassay [14], especially with serum samples, in which botulinum toxin is detected for a shorter time than in feces, particularly in infants [14,51], as well as delays in sample collection, especially feces. For this reason, the confirmed cases of food-borne botulism could be those with more suggestive symptoms and those more serious cases, in which the botulinum toxin is detectable in the serum for a longer time than those with negative results, even if they ate the same contaminated food, explaining a prolonged hospitalization of the confirmed cases. We could not analyze the percentage of positivity among the different samples because, in many cases, only one sample was tested, which is usually serum because it is easier to take and constipation can complicate feces collection. Moreover, in some cases, the information about the nature of samples sent to the laboratory was not collected in the survey or just reflected that with a positive result.

On the other hand, the higher frequency of outbreaks during the summer months explains the higher proportion of probable cases in this season. We decided not to perform a statistical analysis to assess seasonality due to the low number of events, so the test does not have the required statistical power; however, the frequency of cases shows a high difference between seasons. Similarly, in the USA, the three months with the highest number of reported cases are June to August [33]. As this is a retrospective study, we cannot explain the reason for this higher incidence during the third quarter of the year, but we can propose some ideas, such as the difficulty of keeping unrefrigerated preserved food in a cool place and a probable increase in the consumption of non-heated or non-reheated food. After consulting the monthly data for several years on food consumption habits in Spain [52], we could not find an increase in the consumption of home-canned food in summer compared to the rest of the year.

All the commercialized antitoxins for botulism treatment in adults are of an equine origin, which increases the risk of anaphylactic reaction compared to non-equine products [26]. In Spain, no adverse events have been reported through the official pharmacovigilance system, but the online reporting program does not recognize botulinum antitoxin as a drug, forcing manual reporting, with all the classification difficulties and mistakes that this kind of notification entails. None of the cases in our series had diagnostic codes for adverse events to any antitoxin or drug (note that there is not a specific ICD code for botulism antitoxin), but the 72-year-old male case detected in our analysis was also mentioned in a short, previously published article [53]. Therefore, we conclude that, to date, there are no data regarding the frequency of adverse events related to botulism antitoxin in Spain. In our series, we found four cases with confirmed or probable adverse events according to epidemiological surveys and/or an ICD code analysis, giving an anaphylaxis rate of 1.5% and an overall adverse events rate of 3%. These data are in agreement with those published for the current antitoxin (BAT), at around 2.8% and <2% for anaphylaxis [54,55], although all these cases were treated with a previous antitoxin, thus no adverse events have been detected with BAT. However, we did not include some possible adverse events, such as nausea, dizziness or hemodynamic alterations, as it was impossible to distinguish them from the botulism symptoms or the patients’ comorbidities.

The first case of **infant botulism** treated with human antitoxin (BabyBIG) in the European Region took place in Spain in 2007 [56], and, until 2019, it was administered to four more cases without incident. Two other infants were treated with the equine-derived antitoxin with no adverse events reported. To the best of our knowledge, there are few recommendations to prioritize one antitoxin over the other, including in Spain, and we were not able to find any study comparing these two antitoxins. While the CDC recommends human antitoxin because of its safety profile and longer half-life (note that intestinal colonization can last up to three months [57]), the Norwegian Institute of Public Health recommends BAT because it is stocked and less expensive [42,58]. Due to the difficulties in performing a case–control study, we consider it important to share all experiences with both antitoxins. Previous studies detected a reduction in the length of hospitalization in early treated patients [59]. In our series, a delay in antitoxin administration (all the cases were treated >48 h after the onset of symptoms, including one case treated on day +20 [60]), along with a higher level of severity in the treated cases and the low number of total cases, could explain the lack of difference in the duration of hospitalization versus the no-treatment cases. In addition, the absence of infant botulism cases reported in some years is notable and could reflect an under-diagnosis or under-notification.

The absence of wound botulism may suggest a low level of clinical awareness among medical staff. During this period, several wound botulism cases were detected in neighboring countries, including an international outbreak among injecting drug users [27].

The main limitation of the study is the nature of the data collected, which are useful information for the epidemiological and clinical management of each specific case, but which were not collected for a specific study. There are also differences in data collection from different regional public health services and hospitals. As previously mentioned, delays in sample collection notably diminish the chances of getting a confirmed diagnosis, as well as the lack of feces samples, where the toxin can be detected over a longer period than in serum, and typing can be performed. As the majority of the “not specified” toxin type cases probably belong to subtype B, and only 13 cases were related to other toxin types, we did not analyze the differences according to the toxin type, which is a relevant limitation. Another limitation arises from the assumption that survival probabilities are the same for all patients throughout the study period, although it is known that healthcare systems have improved over this time. However, as antitoxin effectiveness is high for all the antitoxins used in this period, is among the untreated cases where differences due to better management in the more recent cases could be significant.

## 4. Conclusions

As a summary of our retrospective study, we conclude that home-canned vegetables remain the most frequent source of food-borne botulism in Spain; the cases involving commercial food are evenly distributed among vegetables, fish and meat products; the outbreaks related to commercialized food are small and scarce, but have still happened in recent years; the fatality rate is as low as in other European countries; the incidence increases during the summer months; and early antitoxin administration notably reduces the length of hospitalization with a very low risk of adverse events. Further studies are needed to assess the risk factors for infection and severity in cases of infant botulism in order to improve its early clinical diagnosis and treatment recommendations. Finally, it is also important to remember the importance of early collection of both feces and food samples (where toxin can be detected for a longer period than in serum) to confirm cases and quickly detect public health risks.

## 5. Material and Methods

This retrospective study was conducted using epidemiological surveillance data from the RENAVE database. To increase the sensitivity and completeness of the data, they were supplemented with the Minimum Basic Data Set (MBDS) of hospital discharge records. Thus, all the cases between 1997 and 2019, registered at hospital discharge and according to the ninth and tenth editions of the International Classification of Diseases (ICD-9 and ICD-10), with any diagnostic of botulism, were reviewed. Due to the low incidence of botulism, it was possible to link each case from the RENAVE database to the MBDS database using basic demographic information (date of birth, sex and place of residence) and the date and place of hospitalization. All the non-coincident data were requested from the regional public health services for review and, in cases of a mismatch, the information from epidemiological surveys (RENAVE) was prioritized. The cases without epidemiological survey data (absence in the RENAVE database) that could not be confirmed by regional public health authorities were excluded. The latest European case definition [61] (the same as the national one) was used to classify the confirmed and probable cases, and the national definition for the possible cases [25] was included (Table 1). Only cases with a positive result in the clinical samples were considered confirmed; for this purpose, serum, prior to antitoxin treatment, and feces samples were collected and analyzed using the standard methods (Table 1). The cases that did not meet the criteria for classification were excluded. The cases requiring ICU admission, ventilation or death were considered severe. The length of hospitalization was calculated as days between admission and discharge dates, collapsing consecutive stays (≤2 day between the first discharge and the next admission date) and transfers into a single episode. For food-borne cases, the incubation period was established as days between the date of consumption of the suspected food and the onset of symptoms.

The ICD codes of other pathologies compatible by clinical symptoms, and often part of the differential diagnosis, were searched to assess the specificity of the included cases in the final dataset. These differential diagnoses included myasthenia gravis, Guillain–Barré and Miller Fisher syndrome, Lamber–Eaton syndrome, stroke-related pathology, poliomyelitis, Wernicke encephalopathy, methanol poisoning and tick-borne encephalitis.

An epidemiological description of the cases was performed, assessing the differences in case classification related to categorical variables using the chi-square test. Logistic regression was used to assess severity by sex, age, whether or not they received antitoxin and the delay until its administration and, for the outbreak-related cases, whether it was the first case detected or a subsequent case. For time series analysis, the frequency of cases per year and the quarter of year were calculated.

To assess the differences in the length of hospitalization based on the time of antitoxin administration, the Cox proportional hazards test and Kruskal–Wallis test were used after checking for non-normality of the data (skewness and kurtosis tests for normality of the length of hospitalization *p*-value= 0.0051). The time to antitoxin administration was classified according to the clinical guideline recommendations [36] in three groups: “early administration” when used within the first 48 h after symptom onset, “later administration” when administered more than 48 h after symptoms onset, and “non-administration” when it was clear that the patient did not receive the antitoxin treatment. Those cases who received antitoxin, but the date was not specified, and were admitted ≥72 h after the symptoms onset were classified in the “later administration” group. We used 72 h instead of 48 h as the breakpoint in order to avoid the misclassification of cases that may have been treated in the emergency ward before getting a hospital admission, even if using 48 h as the breakpoint found similar results. We consider planned discharge as the endpoint and right-censored the observations from the rest (including death, discharge against medical advice, transfer to other unit or center for further management and unknown reason for discharge). The cases receiving an antitoxin that did not match the toxin detected in the clinical samples were considered untreated [36]. When no toxin was detected in the clinical samples or typing was not performed, but patients received an antitoxin, they were considered treated, as the likelihood of matching is higher.

The following symptoms and syndromes related to probable adverse events were searched: urticarial, anaphylaxis, iatrogenic hypotension, allergic angioedema and any antitoxin poisoning. This information was complemented with the epidemiological surveys and public data from the Spanish Medicines and Medical Products Agency (AEMPS, by its acronym in Spanish) which manages the system for drug adverse events notification in Spain. Long-term adverse events were not assessed due to the nature of the available information.

The epidemiological analyses were performed using Stata^®^BE17 and Excell 2016. The results with two-sided *p*-values < 0.05 were considered statistically significant.

## Figures and Tables

**Figure 1 toxins-15-00002-f001:**
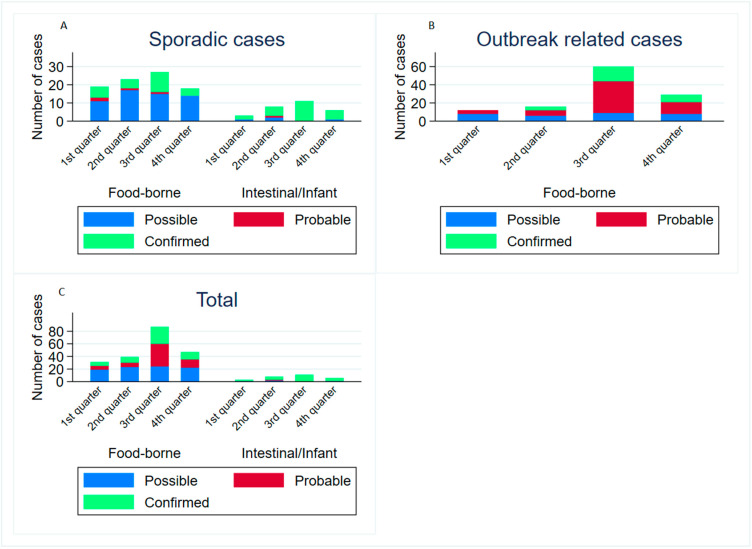
Number of botulism cases per quarter of the year, case classification, disease category and relation with outbreaks, in Spain 1997–2019. (**A**): Sporadic cases (unrelated, single); (**B**): Outbreak related cases; (**C**): Total cases. Note: There was not enough statistical power to obtain statistical significance from its distribution.

**Figure 2 toxins-15-00002-f002:**
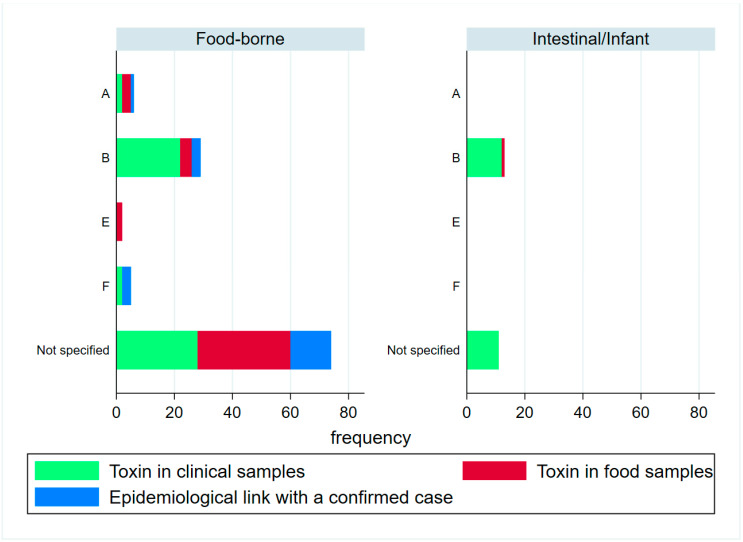
Frequency of botulism cases by toxin type and case classification, in Spain 1997–2019. Note: “Not specified” includes all cases in which the toxin was detected but the type was not identified. Confirmed cases are those in which toxin was detected in clinical samples (green), while probable cases include those in which toxin was detected in food (red) and those involved in an outbreak with confirmed cases (blue). Cases with toxin detection in both clinical and food samples are shown in the “toxin in clinical samples” category.

**Figure 3 toxins-15-00002-f003:**
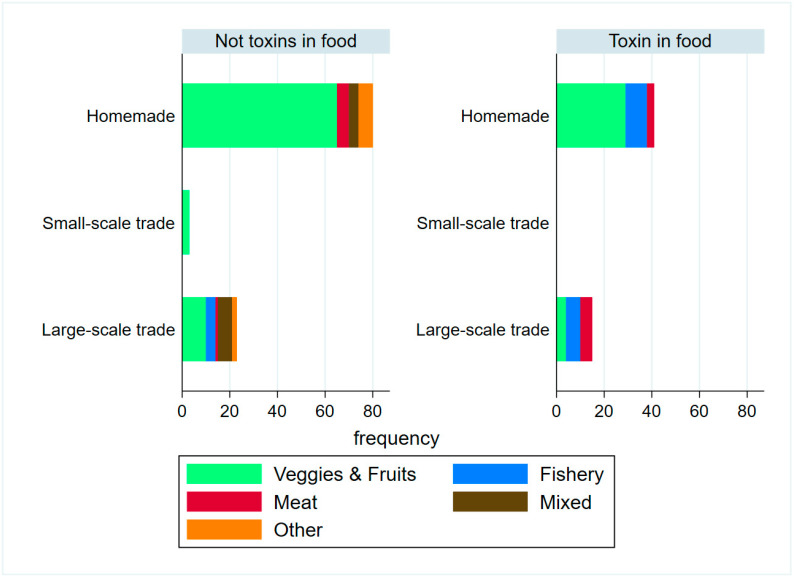
Frequency of food-borne botulism cases by detection of toxins in food, food category and trade-type among possible, probable and confirmed botulism cases in Spain 1997–2019.

**Figure 4 toxins-15-00002-f004:**
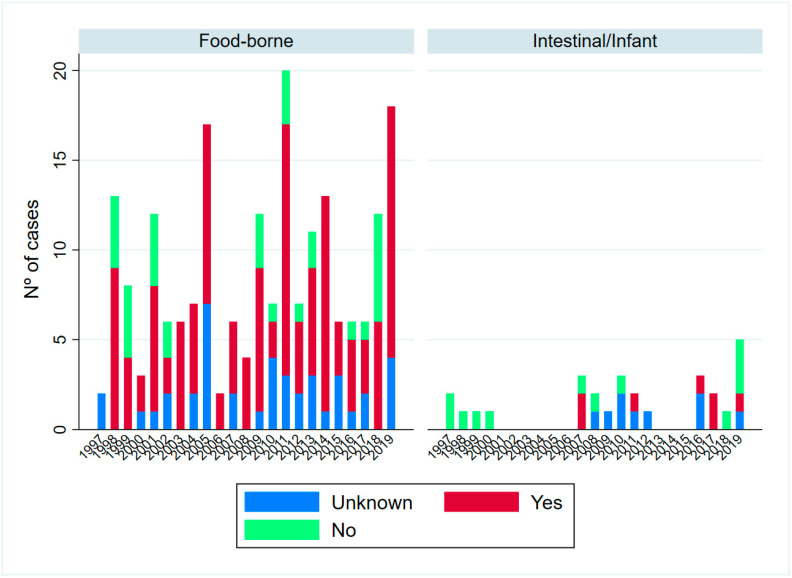
Botulism case distribution by antitoxin treatment, botulism category and year, in Spain 1997–2019.

**Table 1 toxins-15-00002-t001:** Botulism case definitions used to review all cases occurring in Spain from 1997 to 2019.

Case Category	Case Definition
Possible ^	Those meeting clinical criteria and a request for laboratory diagnosis. *Clinical criteria in food-borne cases* comprise at least one of the following two: bilateral cranial nerve impairment and/or peripheral symmetric paralysis. *Clinical criteria in intestinal/infant cases* comprise at least one of the following six: constipation, lethargy, difficulty in sucking or feeding, ptosis; dysphagia and/or general muscle weakness.
Probable ᵟ	Those meeting the clinical criteria and had been exposed to a contaminated food (confirmed by the laboratory) or had an epidemiological link with a confirmed case (related to consumption of the same food as a confirmed case). To detect the botulinum toxin food, the standard mouse bioassay (SMB) was used. The isolation of toxin-producing clostridia in food was used for infant botulism.
Confirmed ᵟ	Those meeting the clinical criteria and with a positive result in the clinical samples. SMB was performed on feces as the preferred sample [30], but standard culture methods and a described multiplex PCR method able to detect A, B, E and F neurotoxin genes in feces, were also carried out [30].
Outbreak	≥2 cases (independent of their category) with an epidemiological link.

ᵟ European and national definition. ^ National definition.

**Table 2 toxins-15-00002-t002:** General characteristics of botulism cases by classification (possible, probable and confirmed) in Spain 1997–2019.

	Total	Confirmed	Probable	Possible	
		N (%)	N (%)	N (%)	
Age group and sex	232	77 (33.19)	63 (27.16)	92 (39.66)	
<1 year (F)	12	11 (91.67)	0	1 (8.33)	
<1 year (M)	18	13 (72.22)	1 (5.56)	4 (22.22)	
1–14 years (F)	3	2 (66.67)	1 (33.33)	0	
1–14 years (M)	8	2 (25)	1 (12.5)	5 (62.5)	
15–24 years (F)	8	1 (12.5)	5 (62.5)	2 (25)	
15–24 years (M)	7	1 (14.29)	2 (28.57)	4 (57.14)	
25–44 years (F)	15	3 (20)	7 (46.67)	5 (33.33)	
25–44 years (M)	32	7 (21.88)	12 (29.27)	18 (43.9)	
45–64 years (F)	31	8 (25.81)	14 (45.16)	9 (29.03)	
45–64 years (M)	41	11 (26.81)	12 (29.27)	18 (43.9)	
65–80 years (F)	30	5 (16.67)	5 (16.67)	20 (66.67)	
65–80 years (M)	22	10 (45.45)	3 (13.64)	9 (40.91)	
>80 years (F)	3	2 (66.67)	0	1 (33.33)	
>80 years (M)	2	1 (50)	0	1 (50)	
*p*-value					<0.001
Category					
Food-borne	204	54 (26.47)	62 (30.39)	88 (43.14)	
Intestinal/infant	28	23 (82.14)	1 (3.57)	4 (14.29)	
*p*-value					<0.001
Outbreak-related					
No	115	49 (42.61)	5 (4.35)	61 (53.04)	
Yes	117	28 (23.93)	58 (49.57)	31 (26.50)	
*p*-value					<0.001
Antitoxin administration					
Yes	133	40 (30.08)	31 (23.31)	62 (46.62)	
No	58	24 (41.38)	14 (24.14)	20 (34.48)	
Unknown	41	13 (31.71)	18 (43.90)	10 (24.39)	
*p*-value					0.025

F: female; M: male.

**Table 3 toxins-15-00002-t003:** Percentage of cases hospitalized within 48 h of the onset of symptoms, mean and median length of hospitalization (in days) for food-borne and infant botulism, in Spain 1997–2019.

	Food-Borne Botulism	Infant Botulism
≤48 h between onset of symptoms and admission	45.20%	44%
Mean hospitalization length (days)	25.56 (SD: 53.7)	21.19 (SD: 10.6)
Median hospitalization length (days)	14 (IQR:18)	19.5 (IQR:17)

**Table 4 toxins-15-00002-t004:** Length of hospitalization analysis by Cox proportional hazards test, and Kruskal–Wallis test, among food-borne botulism cases in Spain 1997–2019.

	Adjusted Analysis (Cox Regression)			Crude Analysis (Kruskal–Wallis)	
	Hazard Ratio	95%CI	*p*-Value	χ² (df) with Ties	*p*-Value
Antitoxin administration category (hours after onset of symptoms)					
≤48 h	1	-	-	5.940 (2)	0.0513
>48 h	0.446	0.219–0.909	0.026	6.219 (1)	0.0126
No antitoxin	0.264	0.116–0.601	0.002	2.695 (1)	0.1007
Age	0.988	0.978–0.998	0.022		
Sex					
Male	1	-	-	0.8998 (1)	0.8998
Female	1.436	0.887–2.327	0.141		
Case classification					
Possible	1	-	-	10.473 (2)	0.0053
Probable	0.783	0.461–1.330	0.365	0.093 (1)	0.7605
Confirmed	0.495	0.281–0.869	0.014	6.903 (1)	0.0086

CI: confidence interval, χ²: chi-square statistic, df: degrees of freedom. For Kruskal–Wallis analysis with one degree of freedom, each category was compared with the previous one (<48 h vs. >48 h, and >48 h vs. no antitoxin administration).

## Data Availability

The data access policy within the National Epidemiological Surveillance Network (RENAVE) is similar to that of other public health agencies, such as the European Centre for Disease Control. The RENAVE, managed and maintained by the National Centre of Epidemiology, has the mandate to collect, analyze and disseminate surveillance data on infectious diseases in Spain. There is not direct access to the RENAVE database, but data are available upon reasonable request to the corresponding author.

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
