# Peer review of "Botulism in Spain: Epidemiology and Outcomes of Antitoxin Treatment, 1997–2019"

_toxins, 2022, doi:10.3390/toxins15010002_

Round 1
Reviewer 1 Report
The information this article is interesting and worth publishing, but the article as it is now needs a lot of improvement.
* As a whole, the article is very confusing as both types of botulism (foodborne and infant) are discussed together, sometimes the reader is not sure to which type of botulism the data or info in the text refers to. These are 2 very different types of botulism with a different clinical diagnosis approach and with different samples to test and different assays that offer the best chance of recovering the organism or the neurotoxin and therefore of confirming a clinical diagnosis: faeces or rectal wash out for infant botulism (serum has little value here), and serum/ faeces/ food for foodborne cases. The source of infection is also very different. While in foodborne botulism, the suspected food is one that can allow the germination of spores and growth of the microorganism and therefore production of toxin, with infant botulism cases, suspected food or environmental samples that may contain spores ie honey, herbal medicine, dust, are investigated and extremely difficult to process and with a much lower chance of detecting C. botulinum. The onset of illness, while it is quite clear in cases of foodborne botulism cases (intoxication), is not easy to determine in infant botulism as this is a toxico infection due to the gut colonisation by C. botulinum and the release of toxin into the blood stream does not occur in a regular manner. Clinical diagnosis for both is not easy and other illness can mimic botulism (Miller Fisher, myasthenia Gravis etc..), therefore, in my opinion, it is best to separate data of both types of botulism as they are not comparable.
* The authors also include statistical analyses of those cases which haven't been microbiologically confirmed (either foodborne of infant, it is not clear when). The authors should take into account that these patients might not have had botulism, and this would explain how some cases (I can't find the data in the text) recovered without antitoxin treatment. These might be some of the clue to suggest this: 'there were no difference (statistically?) between those who received antitoxin and those who did not in terms of length of hospitalisation' L215 (it is correct that maybe the antitoxin treatment had no effect if the treatment occurred too late after onset - but there is no way to be sure whether that is the case or if the illness is not botulism); 'In addition, probability of discharge in length of time (meaning length of stay in hospital?) is lower for confirmed cases than it is possible case' L196, or a page later, about infant botulism: 'There were no differences between those who received antitoxin and those who did not in terms of length of hospitalisation' L215. - infant botulism is a colonisation of the gut by C. botulinum, producing toxin, if the child doesn't received antitoxin, that child will be in hospital for a long time and eventually die. We had one mild case with Type E (C. butyricum), but eventually the child had a relapse, its health degraded and had to be administered with BabyBIG. Botulism toxin is the more potent natural toxin that exist, in all our experience with investigating cases, except for infant botulism, the clinical diagnosis was clear and micro confirmation always eventually happened. In our cases of infant botulism, the clinical diagnosis was not always clear, as expected, but C. botulinum was always recovered from the stool of the infant, as this is a gut colonisation and the organism in our experience can be shed from the host for up to 6 months!
* I agree that Data from unconfirmed case should be recorded but separately: It would be useful to find out why they were unconfirmed. Lack of samples, samples taken too late after onset (for foodborne). Clinical diagnostic not clear etc..
* The authors use terms that are not very accurate in their meaning such as 'positive results' or 'there are no difference between onset of symptoms and hospital admission' L138, 'obtaining discharge', 'toxin type was missing in more than 50% of cases' L236 '(not sure if this is infant of foodborne, but does this mean that toxin was not detected or that the toxin type could not be identified by neutralisation test?) , 'the source was found positive L242, etc...
* There are too many tables and figures, and they are not self-explanatory. They should be. They are quite difficult to read as the headings are not always helpful.
I think table 1 should separate the criteria for foodborne and infant botulism. If only C. botulinum is detected in food, it doesn't mean that that food is responsible for the foodborne botulism cases - if C. botulinum toxin is detected, then yes, it does. If C. botulinum is detected in the stool of an infant, then this is all it takes for confirming a case, together with clinical symptoms. Finding C. botulinum in food stuff used by the baby, like honey or other, when C. botulinum hasn't been detected from the infant stool, doesn't automatically mean that the stuff caused illness.
I don't think table 2 is useful.
Table 3: N/A; does this mean that no food was available for testing?
Figure 1: if this includes non confirmed cases, I don't think it is relevant.
Figure 2: this fig is not clear about what the authors want to show. What does not specified mean? Does it mean that toxin was detected but neutralisation test not possible?
* Absence of cases of wound botulism is not discussed.
Author Response
Thank you very much for your comments and suggestions. We appreciate the time and effort the reviewer has taken to provide all of them. Please see the attachment.

Reviewer 2 Report
This manuscript reports comprehensive epidemiological analysis of botulism in Spain from 1997 and 2019. This well thought work and well written article is all the more interesting since, to my knowledge, there are few examples of such reports that discuss so many information on an extended period : sources of intoxication, serotypes and length of hospitalization … Together these data give valuable information on dominant serotype in Spain, the source of toxi-infections, as well as the impact of the treatment on length of stay in intensive care unit.
This manuscript consists of three parts: introduction, results and discussion. The raw data are presented in the form of seven tables providing definitions of the different categories of cases (T1); distribution by age, category, outbreaks versus sporadic, administration of therapeutics (T2); a table on the characteristics of outbreaks that gives valuable information on the source of contamination (T3); time between onset of symptoms and length of hospitalization that reports a significant reduction of the time period of hospitalization for patients treated before 48h after onset of symptoms with BAT together with statically data on the analysis of the length of hospitalization (T4+5); A table 6 describing acute adverse effects (that is not so necessary to my opinion, while it would be interesting to document here whether or not they were treated with the ABE Behring/BAT, Finely, table 7 summarizes information related to infant botulism.
There are 4 additional figures: F1 showing the distribution of cases related to annual periods (please indicate here that there is not enough statistical power to conclude about a higher rate of contamination during the summer period, ns: for not significant and statistical analysis). F2: frequency by toxin types showing a higher prevalence of serotypes B; F3: known origin of food-borne botulism showing a higher prevalence of disease caused by veggies and fruits and F4 about botulism cases distribution by antitoxin treatment.
The authors also discuss the limits of their study, an interesting aspect in terms of thinking about public health organization across a country.
I enjoyed al lot reading this manuscript. To my opinion it is an interesting work reporting collected data, their thorough analysis and beneficial impact of BAT treatment. I added a few points below that the authors may consider for revision.
Comments:
°Percentages. I have verified percentages in tables and text. You may consider checking lane 84 _ 39.66% (T2); Lane_130 29.88%; lane131_ 65.2 versus 65.5%.
Also, please verify lane 105 (1.9 fold versus 2 )
Table 4 you should think about a better description of the significance of percentages that are calculated within Food-borne and infant patient groups, indeed 45.2+44 is not equal to 100 (not so evident at first time).
If possible, in Table 1 make a subdivision of age and sex because European data show a higher incidence in men than women in two age groups (15-24 and 45-64, 2015) which would be interesting to compare with data presented in your review.
Lane 190: 88 squares out of 103 food-borne versus of 87 food-borne in table4 (see lane 123 is it ok?).
Lane 197: can you give information about the time period if time is not possible?
Lane 200: can you give more precisions regarding time of hospitalization for serotype B only before and after 48h ? and with other unknown serotypes or is it an information that in meaningless considering that not identified BoNT are likely of the B serotype?
Please provide more accurate references instead of Ref9
Please indicate if there is a statistical difference for the two different times of discharge between >48 and no treatment (Lane 190-203, I am not sure this is discussed in the results section).
Author Response

(The authors gave the same response as above.)

Reviewer 3 Report
I read the MS entitled "Botulism in Spain: epidemiology and outcomes of antitoxin treatment, 1997-2019". This retrospective study attempts to provide a more complete picture of the epidemiology of botulism in Spain from 1997 to 2019, and analysing the relationship between delay of antitoxin treatment and hospitalization and the frequency of adverse effects.
The paper is well written and logically structured. It gived the reader a clear picture of the disease in Spain as well an useful knowledge of botulism epidemiology and treatment data from a well known and organised health system. This permits to focus the attention on the disease itself, as well as to the treatment pros and cons. Nevertheless, authors are aware that some data must be considered with a particular attention due to the nature of the data collected which are not part of a specific study or are collected from different regional Public Health services and hospitals.
In conclusion I think that this MS deserves to be published in Toxin after minor revision (see below)
MInor concerns: I suggest to improve graphs. Some are small in my opinion: Fig 1 A can be increased in size by reducing Y range ? Fig 1 B is another panel, so it can have its Y specific range.
Can Fig 2 and 3 be ameliorated by using a PIE graph ? Or changing the colors with a brighter ones ?
Some problems with Fig 4. It seems to me too confused due to the amount of data. Maybe changing with brighter colors ? Or splitting in 2 decades panels (1997-2008)(2009-2019) ?
Author Response

(The authors gave the same response as above.)

Reviewer 4 Report
In this retrospective study, the authors provide a comprehensive epidemiology of botulism in Spain between 1997 to 2019 assessing which treatment was used and, the length of hospitalization in relationship with the injection of antitoxin.
Minor points should be addressed before publication:
Authors should describe in more details BoNTs cellular mechanisms of action, e.g. L chain translocation assisted by the host chaperone Hsp90 and disulphide bridge reduction guarantees by the Trx/TrxR system. Please refer to original publications: Pirazzini M. et al., Cell Reports, 2014 and Azarnia Tehran et al., Cellular Microbiology 2019.
More recent reviews and articles on serotypes classification and botulinum neurotoxins mechanism of action should be citied and discussed. Please take in consideration that, not only chimera, but more than seven serotypes of BoNTs have been discovered. Please refer to: (i) Zornetta I. et al., Scientific Reports, 2016; (ii) Azarnia Tehran D. et al., Toxins 2018; (iii) Zhang et al., Nature Comm., 2017.
Author Response
Thank you very much for your comments and suggestions. Please see the attachment.
